# Research

crystallography/physical chemistry/spectroscopy

methanesulfonate, inelastic neutron scattering spectroscopy, infrared spectroscopy, Raman spectroscopy, density functional perturbation theory

**Author for correspondence:**
Stewart F. Parker
e-mail: stewart.parker@stfc.ac.uk

# Structure and vibrational spectroscopy of lithium and potassium methanesulfonates

Stewart F. Parker[1], Emilie J. Revill-Hivet[2], Daniel W. Nye[1] and Matthias J. Gutmann[1]

[1]ISIS Facility, STFC Rutherford Appleton Laboratory, Chilton, Didcot, Oxon OX11 0QX, UK
[2]Europa School UK, Thame Lane, Culham OX14 3DZ, UK

SFP, 0000-0002-3228-2570

In this work, we have determined the structures of lithium methanesulfonate, $Li(CH_3SO_3)$, and potassium methanesulfonate, $K(CH_3SO_3)$, and analysed their vibrational spectra. The lithium salt crystallizes in the monoclinic space group $C2/m$ with two formula units in the primitive cell. The potassium salt is more complex, crystallizing in $I4/m$ with 12 formula units in the primitive cell. The lithium ion is fourfold coordinated in a distorted tetrahedron, while the potassium salt exhibits three types of coordination: six-, seven- and ninefold. Vibrational spectroscopy of the compounds (including the $^6Li$ and $^7Li$ isotopomers) confirms that the correlation previously found, that in the infrared spectra there is a clear distinction between coordinated and not coordinated forms of the methanesulfonate ion, is also valid here. The lithium salt shows a clear splitting of the asymmetric S–O stretch mode, indicating a bonding interaction, while there is no splitting in the spectrum of the potassium salt, consistent with a purely ionic material.

## 1. Introduction

Derivatives of methanesulfonic acid, $CH_3SO_3H$, which are also known as mesylates, occur widely in chemistry as esters or salts. Some of the organic derivatives are important biologically. This arises because mesylate is a good leaving group in nucleophilic substitution reactions as a result of the efficient delocalization of negative charge between the three oxygen atoms. Thus methyl- and ethylmethanesulfonate are DNA alkylating agents and have been used for many years as DNA damaging agents to induce mutagenesis and in recombination experiments [1,2]. Busulfan (1,4-butanediol dimethanesulfonate) has been used to treat chronic myeloid leukaemia [3].

Metal methanesulfonate salts (M[CH$_3$SO$_3$]$_x$, e.g. M = Na, K, Mg, Ca) occur naturally via the oxidation of dimethyl sulfide and subsequent reaction with the cations present in the ocean [4]. These may then act as condensation nuclei for clouds [5,6]. The alkali metal salts find use in a variety of applications. The potassium salt is used in studies of potassium channels in cells [7] and has been proposed as a novel eluent for liquid chromatography of oligosaccharides [8]. The lithium salt has been tested in a variety of Li-ion batteries [9] because it offers a more stable alternative to the LiPF$_6$ presently used in lithium batteries [10].

We have previously investigated the vibrational spectroscopy of the parent acid, methanesulfonic acid [11] and some of its salts, M = Na, Cs, Cu, Ag, Cd [12]. In the course of our previous work, we have observed a correlation between the type of bonding (ionic or complexed) present and the asymmetric S–O stretch mode in the infrared spectrum. In the present study, we examine the lithium and potassium methanesulfonate salts to further test the correlation. As a prerequisite to this, we have also determined the crystal structures of the compounds.

# 2. Experimental

## 2.1. Materials

K(CH$_3$SO$_3$) (98%), CH$_3$SO$_3$H (99%), $^6$Li$_2$CO$_3$ (95% $^6$Li) and $^7$Li$_2$CO$_3$ (99% $^7$Li) were purchased from Aldrich and used as received. $^6$Li(CH$_3$SO$_3$) and $^7$Li(CH$_3$SO$_3$) were made by the stoichiometric reaction of methanesulfonic acid with the appropriate carbonate. The carbonate ($^6$Li: 1.81 g, $^7$Li: 1.84 g) was suspended in distilled water and the methanesulfonic acid (4.71 g) added dropwise with continuous stirring. The solution was then evaporated to dryness on a hotplate. The yield was 96%.

## 2.2. X-ray crystallography

Single crystal X-ray diffraction data were collected from suitable crystals at 150 K with the Mo K$\alpha$ wavelength using a Rigaku Oxford diffraction Xtalab Synergy S instrument equipped with a liquid nitrogen stream and hybrid pixel array detector (HyPix). The JANA2006 software was used to solve the crystal structure using the built-in charge-flipping algorithm [13]. Details of the refinement are given in table 1 and the CIF files have been deposited with the Cambridge Structural Database. No evidence of impurity phases was found in either dataset.

## 2.3. Vibrational spectroscopy

Inelastic neutron scattering (INS) spectra were recorded at less than 20 K using TOSCA [14] at ISIS.[1] Infrared spectra were recorded using a Bruker Vertex70 FTIR spectrometer, over the range 100–4000 cm$^{-1}$ at 4 cm$^{-1}$ resolution with a DLaTGS detector using 64 scans and the Bruker Diamond ATR. The use of the ultra-wide range beamsplitter enabled the entire spectral range to be recorded without the need to change beamsplitters. The spectra have been corrected for the wavelength-dependent variation in path length using the Bruker software. FT-Raman spectra were recorded with a Bruker MultiRam spectrometer using 1064 nm excitation, 4 cm$^{-1}$ resolution, 500 mW laser power and 64 scans. All the infrared and Raman spectra were measured in air at room temperature.

## 2.4. Computational studies

The plane wave pseudopotential-based program CASTEP was used for the calculation of the vibrational transition energies and their intensities [15,16]. The generalized gradient approximation (GGA) Perdew–Burke–Ernzerhof (PBE) functional was used in conjunction with optimized norm-conserving pseudopotentials. The plane-wave cut-off energy was 830 eV. For the Li salt a $4 \times 6 \times 4$ (48 k-points) Monkhorst–Pack grid was used, for the K salt a $8 \times 8 \times 3$ (96 k-points) grid was used. All of the calculations were converged to better than $|0.009|$ eV Å$^{-1}$. After geometry optimization, the vibrational spectra were calculated in the harmonic approximation using density functional perturbation theory (DFT) [17]. This procedure generates the vibrational eigenvalues and eigenvectors, which allows visualization of the modes within Materials Studio[2] and is also the information needed to calculate the

[1]http://www.isis.stfc.ac.uk.

[2]https://3dsbiovia.com/products/collaborative-science/biovia-materials-studio/.

**Table 1.** Crystal data and structure refinement for lithium and potassium methanesulfonates.

| sample | $LiCH_3SO_3$ | $KCH_3SO_3$ |
| --- | --- | --- |
| empirical formula | $CH_3LiO_3S$ | $CH_3KO_3S$ |
| formula weight | 102.0 | 134.2 |
| temperature (K) | 150(2) | 299(4) |
| wavelength (Å) | 0.71073 (Mo K$\alpha$) | 0.71073 (Mo K$\alpha$) |
| crystal system | monoclinic | tetragonal |
| space group | $C2/m$ | $I4/m$ |
| unit cell dimensions | $a = 7.8181(3)$ Å | $a = 22.1326(3)$ Å |
| | $b = 7.4574(3)$ Å | $c = 6.0532(1)$ Å |
| | $c = 6.5288(3)$ Å | |
| | $\beta = 90.17(2)°$ | |
| volume (Å$^3$) | 380.63(3) | 2965.17(8) |
| $Z$ | 4 | 24 |
| density (calculated) (g cm$^{-3}$) | 1.7805 | 1.8036 |
| absorption coefficient (mm$^{-1}$) | 0.678 | 1.37 |
| F(000) | 208 | 1632 |
| crystal size (mm$^3$) | $0.07 \times 0.06 \times 0.02$ | $0.1 \times 0.06 \times 0.04$ |
| theta range for data collection (°) | 3.10–37.34 | 1.84–29.56 |
| index ranges | $-13 \leq h \leq 13$ | $-28 \leq h \leq 22$ |
| | $102 \leq k \leq 12$ | $-28 \leq k \leq 29$ |
| | $10 \leq l \leq 11$ | $-7 \leq l \leq 7$ |
| reflections collected | 8457 | 20 507 |
| independent reflections ($I > 3\sigma(I)$/all) | 936/1018 | 1731/2078 |
| $R$(int) | 0.0336 | 0.0253 |
| absorption correction | empirical | numerical Gauss integration |
| max. and min. transmission | 1.0 and 0.89 | 1.0 and 0.851 |
| refinement method | full-matrix least squares on $F^2$ | full-matrix least squares on $F^2$ |
| data/constraints/parameters | 1018/2/38 | 2078/6/115 |
| goodness-of-fit on $F^2$ ($I > 3\sigma(I)$/all) | 3.05/2.93 | 2.61/2.40 |
| final R-indices ($I > 3\sigma(I)$) | $R_1 = 0.0296$ | $R_1 = 0.0300$ |
| | $wR_2 = 0.0922$ | $wR_2 = 0.0814$ |
| final $R$-indices (all data) | $R_1 = 0.0321$ | $R_1 = 0.0373$ |
| | $wR_2 = 0.0926$ | $wR_2 = 0.0825$ |
| largest diff. peak and hole (e Å$^{-3}$) | 0.86 and 0.37 | 0.56 and $-0.43$ |

INS spectrum using the program ACLIMAX [18]. Transition energies for isotopic species were calculated from the dynamical matrix that is stored in the CASTEP checkpoint file using the PHONONS utility [19]. We emphasize that the transition energies have *not* been scaled.

# 3. Results and discussion

## 3.1. Structural studies

The structures of the lithium and potassium salts of methanesulfonic acid have been previously determined; however, as far as we are aware, neither has been deposited in a recognized database,

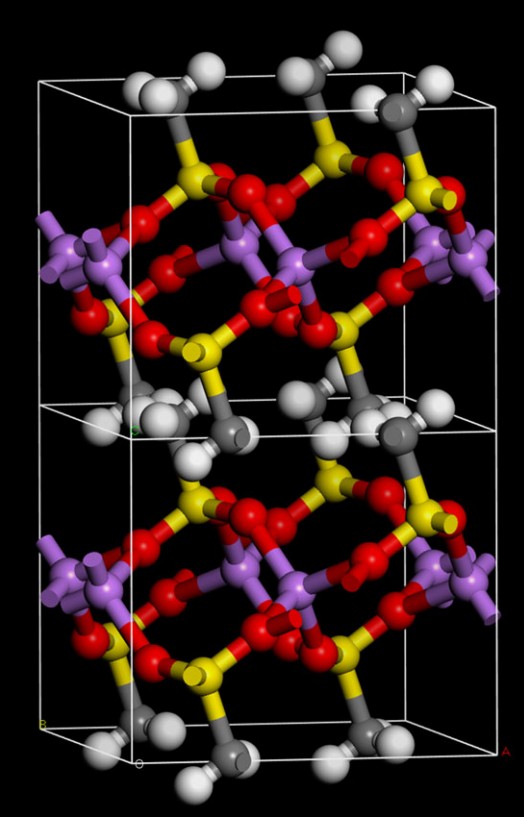

**Figure 1.** Two unit cells of the $C2/m$ structure of Li(CH$_3$SO$_3$). The $c$-axis is vertical. (Grey = carbon, white = hydrogen, red = oxygen, yellow = sulfur, purple = lithium.)

e.g. the Cambridge Structural Database (CSD) [20]. Brief descriptions are provided in conference abstracts (Li [21], K [22]), and the atomic coordinates of the Li salt are given in a thesis [23]; those of the K salt are unavailable. The structure is an essential requirement for the periodic-DFT calculations that we will use to assign the spectra; accordingly, we have re-determined both structures. Table 1 summarizes the results of the structural determinations and figures 1 and 2 show the structures.

Li(CH$_3$SO$_3$) is a relatively simple structure with two formula units arranged centrosymmetrically in the primitive cell. In contrast, K(CH$_3$SO$_3$) is much more complicated with 12 formula units in the primitive cell, comprising three groups of four, each group being on a Wyckoff $h$ site.

Table 2 presents some selected distances. In both structures the methanesulfonate ion lies on a mirror plane, so has $C_s$ symmetry; however, the molecular symmetry is close to $C_{3v}$. Otherwise, the methanesulfonate ion is unremarkable, the molecular geometry is very similar to that found in Na(CH$_3$SO$_3$) [24] and Cs(CH$_3$SO$_3$) [25].

In contrast to the similarity of the methanesulfonate ion in both structures, the coordination of the metal ions is very different: distorted tetrahedral for Li and multiple coordinate for K. On the basis of the infrared spectrum of the Li salt, it had been suggested that the lithium was coordinated to the methanesulfonate [26]. Figure 1 shows that this deduction is correct. Analyses [27,28] of Li–O compounds found that tetrahedral coordination was the most common with <Li–O> = 1.96 Å [27], 1.972 Å [28], completely in accord with that seen here (2 × 1.922, 2 × 2.000 Å). In particular, the Li ion in Li(CF$_3$SO$_3$) [29] shows Li–O distances of 1.873, 1.901, 1.988 and 1.995 Å.

In K(CH$_3$SO$_3$), the potassium ion occupies three distinct sites, with sixfold, sevenfold and ninefold coordination. In each case, the site symmetry is $C_s$. The coordination polyhedra consist of a distorted octahedron, a capped trigonal prism (the cap being on one of the rectangular faces) and a very distorted square antiprism with one of the triangular faces capped. As may be seen in table 3, the K–O distances fall well within the ranges commonly found for the particular type of coordination [28]. Only for sixfold coordination is the average distance seen here apparently somewhat shorter than usually seen, however, the modal K–O distance of 714 structures is 2.72 Å [28], exactly as found here (2.718 Å).

A common motif of the structures of metal methanesulfonates is the separation into polar and non-polar regions. It can be seen from figure 1 that Li(CH$_3$SO$_3$) conforms to this expectation, as it forms a

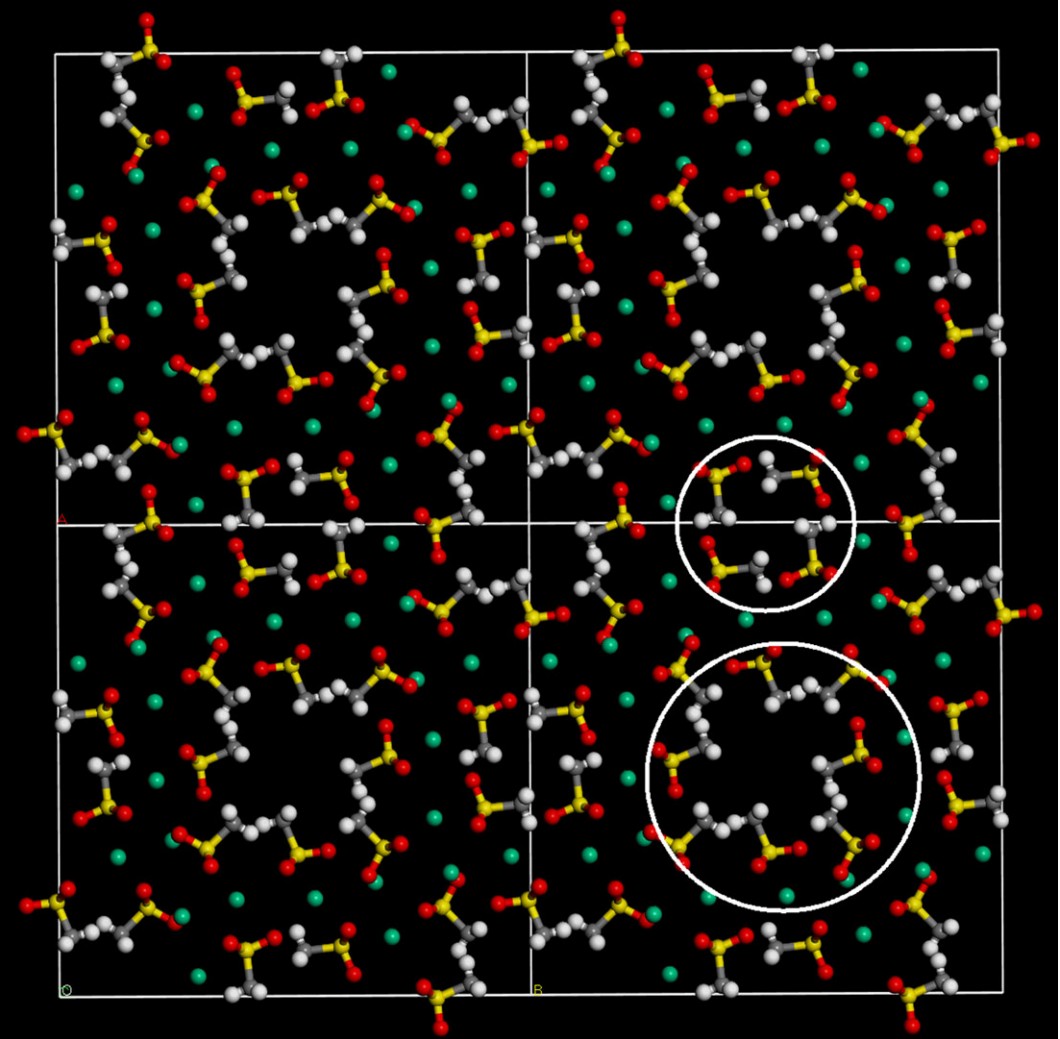

**Figure 2.** Four unit cells of the *I*4/*m* structure of K(CH₃SO₃) viewed along the *c*-axis. (Grey = carbon, white = hydrogen, red = oxygen, yellow = sulfur, green = potassium.)

structure with alternating layers of sulfonate and methyl groups. $K(CH_3SO_3)$ is a much more complex structure; in this case, there are channels running along the *c*-axis that the methyl groups protrude into (highlighted by the large circle in figure 2) with a concentric ring of sulfonate groups and potassium ions. There is an apparent second smaller mixed ring (highlighted by the small circle in figure 2); however, this is deceiving because as figure 3 shows, the methyl and sulfonate groups 'interdigitate' to minimize the interactions.

## 3.2. Vibrational spectroscopy

Figures 4 and 5 show the infrared, Raman and INS spectra of the Li and K salts, respectively. The infrared and Raman spectra of the Li salt [21,23,26] and the infrared spectrum of the K salt [30] have been reported previously. The present spectra are in general agreement with the literature spectra but have an extended transition energy range, and the INS spectra are previously unreported. The spectra of the two salts are broadly similar and do not hint at the complexity of the structure of the K salt. As seen in our previous work [12], the INS spectra are dominated by the methyl modes, particularly the rock (approx. 950 cm$^{-1}$) and the torsion (200–300 cm$^{-1}$). In the K salt, the latter are especially intense. The methyl modes appear only weakly in the infrared and Raman spectra, but they do permit clear observation of the C–H stretch modes that are difficult to see in the INS spectra with this instrument [31]. The infrared and Raman spectra show predominantly the sulfonate modes: S–O stretches (1000–1300 cm$^{-1}$), C–S stretch (approx. 800 cm$^{-1}$), O–S–O bends (500–600 cm$^{-1}$) and the sulfonate rock (approx. 350 cm$^{-1}$). Modes involving significant lithium motion are seen in the range 300–500 cm$^{-1}$ (indicated by * in figure 4).

**Table 2.** Selected bond distances (Å) of lithium and potassium methanesulfonates.

| distance | Li(CH$_3$SO$_3$) | | K(CH$_3$SO$_3$) | |
| --- | --- | --- | --- | --- |
| | observed | calculated | observed | calculated |
| C1–H | 0.939, 2 × 0.848 | 1.095, 2 × 1.094 | 2 × 0.950, 0.978 | 1.095, 2 × 1.096 |
| C2–H | | | 2 × 0.920, 0.934 | 3 × 1.096 |
| C3–H | | | 0.934, 2 × 0.854 | 2 × 1.094, 1.096 |
| C123–S | 1.743 | 1.771 | 1.752, 1.743, 1.756 | 1.783, 1.783, 1.783 |
| S1–0 | 1.443, 2 × 1.471 | 1.485, 2 × 1.469 | 2 × 1.451, 1.452 | 2 × 1.474, 1.477 |
| S2–0 | | | 1.434, 2 × 1.452 | 1.465, 2 × 1.479 |
| S3–0 | | | 1.422, 2 × 1.414 | 1.472, 2 × 1.474 |
| M–0 | 2 × 1.922, 2 × 2.000 | 2 × 1.925, 2 × 1.993 | K1: 2.666, 2 × 2.804, 2 × 2.827, 2 × 2.947, 2 × 3.062 | K1: 2.715, 2 × 2.813, 2 × 2.842, 2 × 2.972, 2 × 3.070 |
| | | | K2: 2.646, 2 × 2.677, 2.712, 2 × 2.799 | K2: 2.680, 2 × 2.702, 2.712, 2 × 2.828 |
| | | | K3: 2 × 2.689, 2 × 2.753, 2 × 2.974, 3.061 | K3: 2 × 2.692, 2 × 2.775, 2 × 2.938, 2.963 |

**Table 3.** The coordination around the K$^+$ ions of potassium methanesulfonate. Short, Long and Ave. are the shortest, longest and average K–0 distances (all in Å).

| coordination number | K(CH$_3$SO$_3$) | | | literature [28] | | |
| --- | --- | --- | --- | --- | --- | --- |
| | Short | Long | Ave. | Short | Long | Ave. |
| 6 | 2.646 | 2.799 | 2.718 | 2.447 | 3.587 | 2.828 |
| 7 | 2.689 | 3.061 | 2.842 | 2.524 | 3.554 | 2.861 |
| 9 | 2.666 | 3.062 | 2.883 | 2.491 | 3.797 | 2.955 |

To provide more definitive assignments requires periodic-DFT calculations. Figure 6 compares the observed and calculated INS spectra of Li(CH$_3$SO$_3$) and K(CH$_3$SO$_3$). It can be seen that the agreement is reasonable in terms of both the transition energy and the relative intensities. This is more so for the Li compound because the calculation is for the entire Brillouin zone, whereas it is for the $\Gamma$-point only for the K compound because of the complexity of the system. The intensity mismatch in the region greater than 800 cm$^{-1}$ is likely to be the result of the Debye–Waller factor being too large because the lattice mode region is calculated to be too strong.

Nonetheless, the agreement is sufficiently good as to allow definitive assignments. Li(CH$_3$SO$_3$) crystallizes in the monoclinic space group $C2/m$ (no. 12) with two formula units in the primitive cell, thus there are 54 modes in total comprising 3 acoustic modes, 9 optic translational modes of the ions, together with 6 librational and 36 internal modes of the methanesulfonate ion. Similarly, K(CH$_3$SO$_3$) crystallizes in the tetragonal space group $I4/m$ (no. 87) with 12 formula units in the primitive cell, thus there are 324 modes in total comprising three acoustic modes, 69 optic translational modes of the

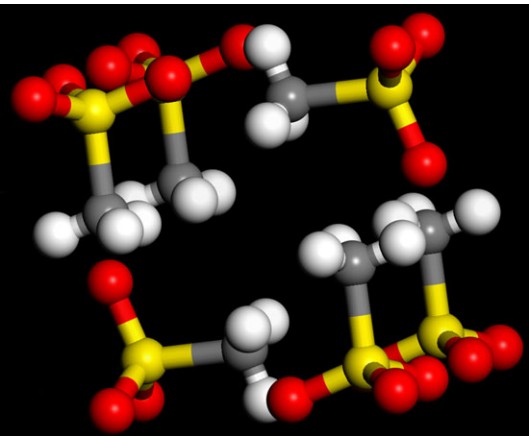

**Figure 3.** Expanded view of the apparent 'mixed' ring in the *I*4/*m* structure of K(CH$_3$SO$_3$). (Grey = carbon, white = hydrogen, red = oxygen, yellow = sulfur, the K$^+$ ions are omitted for clarity.)

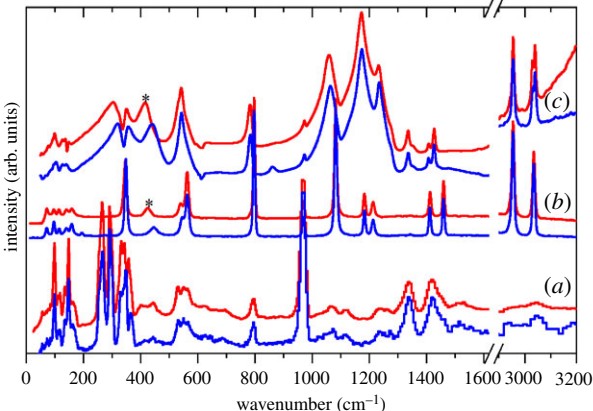

**Figure 4.** Vibrational spectra of Li(CH$_3$SO$_3$): (*a*) INS, (*b*) Raman and (*c*) infrared (the 2900–3200 cm$^{-1}$ is ×5 ordinate expanded relative to the 0–1600 cm$^{-1}$ region). For each pair of spectra, the upper (red) trace is the $^7$Li isotopomer and the lower (blue) trace is the $^6$Li isotopomer. The * indicates Li sensitive modes.

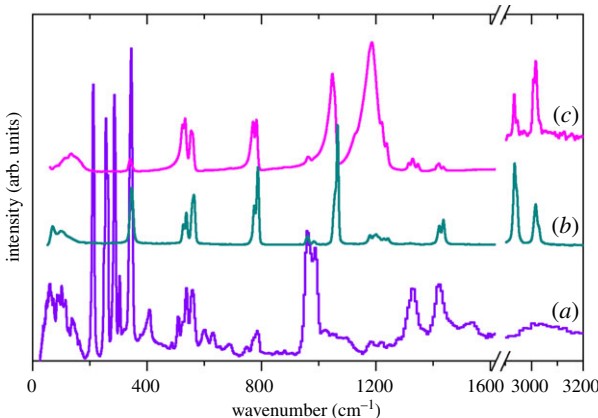

**Figure 5.** Vibrational spectra of K(CH$_3$SO$_3$): (*a*) INS, (*b*) Raman and (*c*) infrared (the 2900–3200 cm$^{-1}$ is ×10 ordinate expanded relative to the 0–1600 cm$^{-1}$ region).

ions, together with 36 librational and 216 internal modes of the methanesulfonate ion. This means that each mode of the 'free' M(CH$_3$SO$_3$) species will give rise to four (Li) or 12 (K) factor group components. Inspection of figures 4 and 5 gives no indication of significant factor group splitting in the spectra, with the exception of the multiple methyl torsions in the K compound, and this

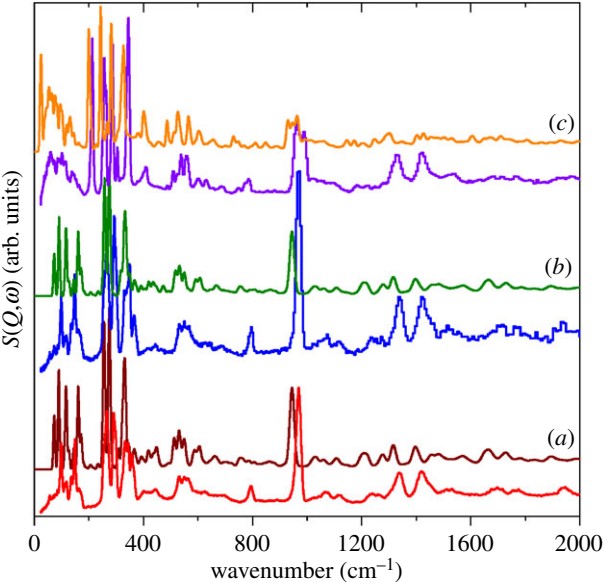

**Figure 6.** Comparison of experimental (red, blue and violet) and calculated (brown, olive and orange) INS spectra of: (a) $^{7}$Li(CH$_3$SO$_3$), (b) $^{6}$Li(CH$_3$SO$_3$) isotopomer and (c) K(CH$_3$SO$_3$).

is confirmed by the calculations. In the K salt, the methanesulfonates occupy three independent Wyckoff $h$ sites and each of these is responsible for one of the torsion modes at 213, 257 and 286 cm$^{-1}$, (the fourth very strong mode at 343 cm$^{-1}$ is a rocking mode of the sulfonate group, which results in a large displacement of the methyl group, accounting for its intensity). Table 4 lists the observed modes and the average of the factor group splitting (except for the torsions) of the calculated modes with their assignments.

As seen previously [12], only the methyl-related modes (C–H bends, rock and torsion), have significant intensity in the INS spectrum and demonstrates that the coupling between the CH$_3$ and SO$_3$ functionalities in the ion is weak. The strongest modes in the infrared and Raman spectra are motions of the sulfonate group, as these involve significant charge distortions that generate the intensity.

As noted earlier, the metal coordination is distinctly different in the two compounds: fourfold for Li and six-, seven- and ninefold for K. The bond distances are also very different: 1.922–2.000 for Li and 2.652–3.222 for K. We take these differences to indicate that the interaction with Li is significantly stronger than for K. The calculated spectra provide support for this idea. Figure 7 shows pseudo-INS spectra calculated by setting the cross section of the atom of interest to 100 barn and all other atoms to 0 barn. Thus only modes that involve motion of the atom will contribute to the spectrum. For the K salt, it can be seen that all the metal ion modes occur below 200 cm$^{-1}$ (figure 7a), while for the Li salt there are two groups of metal ion modes at 300–350 and 400–480 cm$^{-1}$ (figure 7b,c). Inspection of the mode animations shows that the former arise from a coupled motion with the sulfonate rock modes. The latter can be considered to be either Li translations or Li–O bond stretching. In the K salt, the distances are consistent with a purely ionic material, so by calculating the spectrum for the K salt but with a mass of 7 amu, i.e. $^{'7}$K$'$, we approximate what the transition energies would be for a Li ion that is only involved in ionic interactions. The result is shown in figure 7d and it can be seen that the maximum energy is 350 cm$^{-1}$, approximately 100 cm$^{-1}$ below that seen in the Li salt. This suggests that there is an additional interaction in the Li salt, thus the description of the modes as Li–O bond stretching is the better choice.

In previous work [12], we showed that in compounds with coordinated methanesulfonate ions, the asymmetric S–O stretch mode is both strongly perturbed and is downshifted with respect to purely ionic compounds. This is best seen in the infrared spectra and a comparison of the Li and K salts with those studied earlier—Cs(CH$_3$SO$_3$), Na(CH$_3$SO$_3$), Ag(CH$_3$SO$_3$), Cd(H$_2$O)$_2$(CH$_3$SO$_3$)$_2$ and Cu(H$_2$O)$_4$(CH$_3$SO$_3$)$_2$—is shown in figure 8. It can be seen that the degeneracy of the S–O asymmetric stretch at 1100–1250 cm$^{-1}$ is lifted and two modes appear. (For the Cd salt, this manifests as a pronounced broadening of the band.) While the spectrum of the K salt is very similar to that of the Cs and Na salts, the distinct splitting of the S–O asymmetric stretch in the Li salt is reminiscent of that found in the coordination compounds, consistent with Li–O bonding.

**Table 4.** Observed and the average of the calculated factor group splitting (CASTEP) transition energies ($cm^{-1}$) of $^6Li(CH_3SO_3)$, $^7Li(CH_3SO_3)$ and $K(CH_3SO_3)$. (v, very; s, strong; m, medium; w, weak; br, broad; sh, shoulder).

| Li(CH₃SO₃) | | | | K(CH₃SO₃) | | | | |
| --- | --- | --- | --- | --- | --- | --- | --- | --- |
| CASTEP | INS | Raman | Infrared | CASTEP | INS | Raman | Infrared | description |
| 3100 | | | 3040w | 3084 | | 3028sh | 3017w, | CH₃ asymmetric stretch |
| 3099 | | 3034w | 3030w | 3069 | | 3015w | 3007w | CH₃ asymmetric stretch |
| 2990 | | 2955w | 2955w | 2839 | | 2944sh, 2935w | 2934w | CH₃ symmetric stretch |
| 1433 | 1423 m | | 1427w | 1420 | 1420s | 1436w | 1435w | CH₃ asymmetric bend |
| 1396 | | 1412w | 1407w | 1402 | | 1422w | 1421w | CH₃ asymmetric bend |
| 1317 | 1340 m | 1343vw | 1336w | 1300 | 1330s | | 1349, 1330, 1315 | CH₃ symmetric bend |
| 1191 | | 1213w | 1236s | 1192 | 1221w | 1243w, 1228w, | 1238sh, 1222sh, | SO₃ asymmetric stretch |
| 1141 | | 1184w | 1174vs | 1156 | 1181w | 1212sh, 1201w, 1196sh,1180w | 1186vs,br, 1127sh | SO₃ asymmetric stretch |
| 1039 | | 1082s | 1065s | 1022 | | 1066vs, 1058sh | 1048vs | SO₃ symmetric stretch |
| 949 | 970vs | 970w | 973w | 958 | 987s | 983w | | CH₃ rock |
| 941 | | | | 934 | 963s | 961w | 964w | CH₃ rock |
| 757 | 797w | 797s | 783 m | 742 | 785w, 769sh | 788s, 776 m | 783s, 771s | C–S stretch + SO₃ symmetric bend |
| 559 (⁶Li)<br>555 (⁷Li) | 565w (⁶Li)<br>565w (⁷Li) | 564 m (⁶Li)<br>563 m (⁷Li) | | 532 | | 564 m | 560sh, 555 m | SO₃ symmetric bend + C–S stretch |
| 538 (⁶Li)<br>533 (⁷Li) | 551 m (⁶Li)<br>551 m (⁷Li) | 546w (⁶Li)<br>539w (⁷Li) | 543w (⁶Li)<br>542w (⁷Li) | 515 | | 538w | 534s | SO₃ asymmetric bend |
| 520 (⁶Li)<br>515 (⁷Li) | 532w (⁶Li)<br>532w (⁷Li) | | | 503 | | 527w | 524s | SO₃ asymmetric bend |
| 477 (⁶Li)<br>455 (⁷Li) | | 447w (⁶Li)<br>425w (⁷Li) | 441w (⁶Li)<br>416w (⁷Li) | | | | | Li⁺ translation |
| 458 (⁶Li)<br>437 (⁷Li) | | | | | | | | Li⁺ translation |

(Continued.)

**Table 4.** (*Continued.*)

| Li(CH₃SO₃) | | | | K(CH₃SO₃) | | | | |
|---|---|---|---|---|---|---|---|---|
| CASTEP | INS | Raman | Infrared | CASTEP | INS | Raman | Infrared | description |
| 354 ($^6$Li) 335 ($^7$Li) | | | 357 m ($^6$Li) 351 m ($^7$Li) | | | | | Li$^+$ translation |
| 334 ($^6$Li) 330 ($^7$Li) | 349s ($^6$Li) 343s ($^7$Li) | 349 m ($^6$Li) 346 m ($^7$Li) | | 329 | 343vs | 346w | 341s | SO$_3$ rock |
| 323 ($^6$Li) 316 ($^7$Li) | 333s ($^6$Li) 332s ($^7$Li) | | | 321 | | | | SO$_3$ rock |
| 275 | 292s | | | 283 | 286vs | | | CH$_3$ torsion |
| 257 | 266s | | | 244 | 257vs | | | CH$_3$ torsion |
| | | | | 202 | 213vs | | | CH$_3$ torsion |

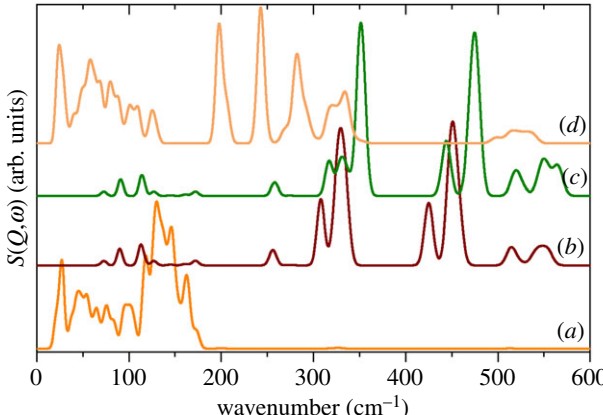

**Figure 7.** Pseudo-INS spectra of the modes that involve metal ion motion. (a) $^{nat}$K(CH$_3$SO$_3$), (b) $^7$Li(CH$_3$SO$_3$), (c) $^6$Li(CH$_3$SO$_3$) and (d) $^{,7}$K(CH$_3$SO$_3$)'.

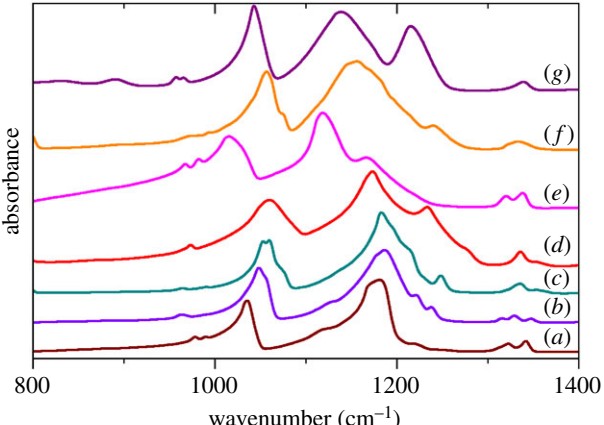

**Figure 8.** Infrared spectra of (a) Cs(CH$_3$SO$_3$), (b) K(CH$_3$SO$_3$), (c) Na(CH$_3$SO$_3$), (d) $^7$Li(CH$_3$SO$_3$), (e) Ag(CH$_3$SO$_3$), (f) Cd(H$_2$O)$_2$(CH$_3$SO$_3$)$_2$ and (g) Cu(H$_2$O)$_4$(CH$_3$SO$_3$)$_2$ in the S–O stretch mode region of the sulfonate ion. The symmetric stretch is at 1000–1050 cm$^{-1}$ and asymmetric stretch is at 1100–1250 cm$^{-1}$.

## 4. Conclusion

In this work, we have determined the structures of lithium and potassium methanesulfonates and analysed their vibrational spectra. The structural study shows that the metal coordination is not unusual, although the presence of three types—six-, seven- and ninefold—in the potassium salt is noteworthy. The vibrational spectroscopy confirms that the correlation previously found [12], that in the infrared spectrum there is a clear distinction between coordinated and not coordinated forms of the methanesulfonate ion, is also valid here. The lithium salt shows a clear splitting of the asymmetric S–O stretch mode, indicating a bonding interaction, while there is no splitting in the spectrum of the potassium salt, consistent with a purely ionic material.

Data accessibility. The datasets supporting this article are available from the Science and Technology Facilities data repository eData at: http://dx.doi.org/10.5286/edata/739. The structures of lithium methanesulfonate and potassium methanesulfonate have also been deposited with the CSD [20]. The deposit numbers are: CCDC 1989314 for K(CH$_3$SO$_3$) and CCDC 1989315 for Li(CH$_3$SO$_3$). The INS spectra of $^6$Li(CH$_3$SO$_3$), $^7$Li(CH$_3$SO$_3$) and K(CH$_3$SO$_3$) are available from the INS database at: http://wwwisis2.isis.rl.ac.uk/INSdatabase/.

Authors' contributions. E.J.R.-H. made the $^6$Li(CH$_3$SO$_3$) and $^7$Li(CH$_3$SO$_3$) salts and measured the infrared and Raman spectra of all the compounds; D.W.N. collected the single-crystal X-ray data; M.J.G. carried out the structure solution; S.F.P. measured the INS spectra, carried out the DFT calculations and wrote the manuscript. All authors gave final approval for publication.

Competing interests. We declare we have no competing interests.

Funding. This work is supported by the Science and Technologies Research Council (STFC).

**Acknowledgements.** The STFC Rutherford Appleton Laboratory is thanked for access to neutron beam facilities. Computing resources (time on the SCARF compute cluster for the CASTEP calculations) was provided by STFC's e-Science facility. This research has been performed with the aid of facilities at the Research Complex at Harwell, including the FT-Raman spectrometer. The authors would like to thank the Research Complex for access to, and support of, these facilities and equipment.

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
