## [Reviewer comments · Royal Society Open Science]

Review History

RSOS-200413.R0 (Original submission)

Review form: Reviewer 1

Is the manuscript scientifically sound in its present form?

Yes

Are the interpretations and conclusions justified by the results?

Yes

Is the language acceptable?

Yes

Do you have any ethical concerns with this paper?

No

Have you any concerns about statistical analyses in this paper?

No

Recommendation?

Accept with minor revision (please list in comments)

Comments to the Author(s)

This is a really nice paper, communicating important vibrational analysis data on methanesulfonate. While I find the paper a bit too short (Introduction and Methods), the data and analysis are very good. I would recommend acceptance with minor corrections.

Review form: Reviewer 2

Is the manuscript scientifically sound in its present form?

Yes

Are the interpretations and conclusions justified by the results?

Yes

Is the language acceptable?

Yes

Do you have any ethical concerns with this paper?

No

Have you any concerns about statistical analyses in this paper?

No

Recommendation?

Major revision is needed (please make suggestions in comments)

Comments to the Author(s)

The authors successfully collected vibrational spectra of Li and K methanesulfonates and observed satisfactory agreement between calculated and observed spectra. Upon this, they demonstrated a fourfold structure for Li salt and multi-fold structure for K salt. But I think the authors should introduce the motivation of this work more clearly and how it is related to the previous investigations.

Beside this, there are some other points that I think need to be addressed:

1. The observation of different coordination modes seems to be interesting. Could the authors discuss more on this?
2. When comparing the Li-O distance with literature values, the authors compared tetrahedral coordinated Li-O. How about K-O since there are different types of coordinated K in this compound?
3. There are a lot of figures and quite a long table, have the authors thought about merge several figures or put some in supplementary information? In addition, the experimental and calculated spectra are overlapped in Figure 6 and are difficult to clearly compared. It may be better to shift the traces a little bit.
4. The authors mentioned for the Li salt they observed distinct patterns. It's not the case for K salt since it's a purely ionic material. Have the authors tried to synthesis K salt and compare?

Review form: Reviewer 3

Is the manuscript scientifically sound in its present form?

No

Are the interpretations and conclusions justified by the results?

Yes

Is the language acceptable?

Yes

Do you have any ethical concerns with this paper?

No

Have you any concerns about statistical analyses in this paper?

No

Recommendation?

Reject

Comments to the Author(s)

Parker et al. report in this manuscript the study on the crystal structures of the lithium and potassium methanesulfonates. Both compounds have been characterised through the inelastic neutron scattering, Raman and infrared spectroscopies. In addition, DFT calculations have been performed. These studies are of high quality, but some doubts arise from this work. The introduction is extremely short, the syntheses of the studied compounds have not been reported in detail (what about the yield of the synthetic procedure and the analysis of the purity and homogeneity of the samples?), and some parts of the crystallographic data have been communicated previously in conference abstracts and in a Thesis (in the case of the Li compound). In my opinion, in its present form, the manuscript is neither suitable nor scientifically sound. For these reasons, I do not recommend its publication in the journal RSOS (maybe this work could be more suitable for Polyhedron or RSC Adv.)

Decision letter (RSOS-200413.R0)

15-Apr-2020

Dear Dr Parker:

Manuscript ID: RSOS-200413

Title: "Structure and vibrational spectroscopy of lithium and potassium methanesulfonates"

Thank you for submitting the above manuscript to Royal Society Open Science. Your paper was sent to reviewers and their comments are included at the bottom of this letter.

In view of the concerns raised by the reviewers, the manuscript has been rejected in its current form. However, a new manuscript may be submitted which takes into consideration these comments.

Please note that resubmitting your manuscript does not guarantee eventual acceptance, and that your resubmission will be subject to peer review before a decision is made.

Your resubmitted manuscript should be submitted by 13-Oct-2020. If you are unable to submit by this date please contact the Editorial Office.

On behalf of the Subject Editor Professor Anthony Stace and the Associate Editor Dr Ya-Wen Wang

REVIEWER(S) REPORTS:
Associate Editor Comments to Author ():
RSC Associate Editor:
Comments to the Author:
(There are no comments.)

RSC Subject Editor:
Comments to the Author:
(There are no comments.)

Reviewers' Comments to Author:
Reviewer: 1

Comments to the Author(s)
This is a really nice paper, communicating important vibrational analysis data on methanesulfonate. While I find the paper a bit too short (Introduction and Methods), the data and analysis are very good. I would recommend acceptance with minor corrections.

Reviewer: 2

Comments to the Author(s)
The authors successfully collected vibrational spectra of Li and K methanesulfonates and observed satisfactory agreement between calculated and observed spectra. Upon this, they demonstrated a fourfold structure for Li salt and multi-fold structure for K salt. But I think the authors should introduce the motivation of this work more clearly and how it is related to the previous investigations.

Beside this, there are some other points that I think need to be addressed:

1. The observation of different coordination modes seems to be interesting. Could the authors discuss more on this?
2. When comparing the Li-O distance with literature values, the authors compared tetrahedral coordinated Li-O. How about K-O since there are different types of coordinated K in this compound?
3. There are a lot of figures and quite a long table, have the authors thought about merge several figures or put some in supplementary information? In addition, the experimental and calculated spectra are overlapped in Figure 6 and are difficult to clearly compared. It may be better to shift the traces a little bit.
4. The authors mentioned for the Li salt they observed distinct patterns. It's not the case for K salt since it's a purely ionic material. Have the authors tried to synthesis K salt and compare?

Reviewer: 3

Comments to the Author(s)

Parker et al. report in this manuscript the study on the crystal structures of the lithium and potassium methanesulfonates. Both compounds have been characterised through the inelastic neutron scattering, Raman and infrared spectroscopies. In addition, DFT calculations have been performed. These studies are of high quality, but some doubts arise from this work. The introduction is extremely short, the syntheses of the studied compounds have not been reported in detail (what about the yield of the synthetic procedure and the analysis of the purity and homogeneity of the samples?), and some parts of the crystallographic data have been communicated previously in conference abstracts and in a Thesis (in the case of the Li compound). In my opinion, in its present form, the manuscript is neither suitable nor scientifically sound. For these reasons, I do not recommend its publication in the journal RSOS.

Author's Response to Decision Letter for (RSOS-200413.R0)

See Appendix A.

RSOS-200776.R0

Review form: Reviewer 2

Is the manuscript scientifically sound in its present form?

Yes

Are the interpretations and conclusions justified by the results?

Yes

Is the language acceptable?

Yes

Do you have any ethical concerns with this paper?

No

Have you any concerns about statistical analyses in this paper?

No

Recommendation?

Accept as is

Comments to the Author(s)

I think the authors make significant efforts to address the issues and the current manuscript is in better shape to be published on RSOS.

Review form: Reviewer 3

Is the manuscript scientifically sound in its present form?

Yes

Are the interpretations and conclusions justified by the results?

Yes

Is the language acceptable?

Yes

Do you have any ethical concerns with this paper?

No

Have you any concerns about statistical analyses in this paper?

No

Recommendation?

Accept with minor revision (please list in comments)

Comments to the Author(s)

I appreciate the efforts that the authors have made in response to my questions and concerns. The revision clarifies almost all the points I raised and helps me (and hopefully readers) to understand this manuscript. I think the authors may still take into account the following point: Please, regarding the yield reported in the revised manuscript, give a more accurate value ("...>90%" is not valid for me).

Decision letter (RSOS-200776.R0)

Dear Dr Parker:

Title: Structure and vibrational spectroscopy of lithium and potassium methanesulfonates
Manuscript ID: RSOS-200776

Thank you for submitting the above manuscript to Royal Society Open Science. On behalf of the Editors and the Royal Society of Chemistry, I am pleased to inform you that your manuscript will be accepted for publication in Royal Society Open Science subject to minor revision in accordance with the referee suggestions. Please find the reviewers' comments at the end of this email.

The reviewers and handling editors have recommended publication, but also suggest some minor revisions to your manuscript. Therefore, I invite you to respond to the comments and revise your manuscript.

Because the schedule for publication is very tight, it is a condition of publication that you submit the revised version of your manuscript before 07-Jun-2020. Please note that the revision deadline will expire at 00.00am on this date. If you do not think you will be able to meet this date please let me know immediately.

Kind regards,

Dr Laura Smith
Publishing Editor, Journals

RSC Associate Editor
Comments to the Author:
(There are no comments.)

Reviewer comments to Author:
Reviewer: 3

Comments to the Author(s)
I appreciate the efforts that the authors have made in response to my questions and concerns. The revision clarifies almost all the points I raised and helps me (and hopefully readers) to understand this manuscript. I think the authors may still take into account the following point: Please, regarding the yield reported in the revised manuscript, give a more accurate value ("...>90%" is not valid for me).

Reviewer: 2

Comments to the Author(s)
I think the authors make significant efforts to address the issues and the current manuscript is in better shape to be published on RSOS.

Author's Response to Decision Letter for (RSOS-200776.R0)

See Appendix B.

Decision letter (RSOS-200776.R1)

Dear Dr Parker:

Title: Structure and vibrational spectroscopy of lithium and potassium methanesulfonates
Manuscript ID: RSOS-200776.R1

It is a pleasure to accept your manuscript in its current form for publication in Royal Society Open Science. The chemistry content of Royal Society Open Science is published in collaboration with the Royal Society of Chemistry.

RSC Associate Editor
Comments to the Author:
(There are no comments.)

Reviewer(s)' Comments to Author:

Appendix A

ISIS Facility

Science and Technology Facilities Council
Rutherford Appleton Laboratory, Harwell Science and
Innovation Campus, Didcot, OX11 0QX United Kingdom

www.scitech.ac.uk

5th May 2020

Prof Stewart F. Parker
ISIS Facility
STFC Rutherford Appleton Laboratory
Chilton, Didcot
Oxfordshire
OX11 0QX
United Kingdom

Direct line +44 (0)1235 6182
E-mail stewart.parker@stfc.ac.uk

**“Structure and vibrational spectroscopy of lithium and potassium methanesulfonates”
(Manuscript ID: RSOS-200413)**

Dear Dr Smith,

Thank you for the reviewers comments, I see that we have the full gamut from “accept as is” to “minor revision” to “reject”! On the following pages, I give detailed responses to the points raised by the reviewers. I hope that with these changes the manuscript is now acceptable for publication in *Royal Society Open Science*.

Yours sincerely

Prof Stewart F. Parker

In the following we give the original comment in plain text and *our response in italics*. Changes in the manuscript are highlighted in yellow in the file:

“Parker_Li_&_K_methanesulfonates_RSOS_Revision.docx”.

Reviewer: 1

Comments to the Author(s)

This is a really nice paper, communicating important vibrational analysis data on methanesulfonate. While I find the paper a bit too short (Introduction and Methods), the data and analysis are very good. I would recommend acceptance with minor corrections.

We thank the reviewer for their support. All three reviewers commented that they considered the Introduction to be too short. In the Introduction of the revised manuscript, we have expanded on the motivation for the paper and its relationship to previous work.

Reviewer: 2

Comments to the Author(s)

The authors successfully collected vibrational spectra of Li and K methanesulfonates and observed satisfactory agreement between calculated and observed spectra. Upon this, they demonstrated a four-fold structure for Li salt and multi-fold structure for K salt. But I think the authors should introduce the motivation of this work more clearly and how it is related to the previous investigations.

As noted above, we have expanded on the motivation for the paper and its relationship to previous work.

Beside this, there are some other points that I think need to be addressed:

1. The observation of different coordination modes seems to be interesting. Could the authors discuss more on this?

We believe that this is already included in the paper, particularly in section 3.1 Structural studies.

2. When comparing the Li-O distance with literature values, the authors compared tetrahedral coordinated Li-O. How about K-O since there are different types of coordinated K in this compound?

In the revised manuscript, we discuss the coordination around the potassium ion in more detail.

The details of Table 2 regarding the distances in the K salt have been updated as the table was constructed from an early version of the structure before the refinement was completed. We also realised on examination of the final structure that some of the distances assigned as K–O were actually K·H, this means that the coordination numbers around K are 6, 7 and 9, not 7, 8 and 9 as originally stated.

3. There are a lot of figures and quite a long table, have the authors thought about merge several figures or put some in supplementary information? In addition, the experimental and calculated spectra are overlapped in Figure 6 and are difficult to clearly compared. It may be better to shift the traces a little bit.

We have considered whether it would be possible to combine some of the figures. The only ones that this would be reasonable for are 4 with 5 and 6 with 7. Combining 4 with 5 would result in a very complicated figure, that would be difficult to follow. In addition, Figure 5 allows the extraordinary strength of the methyl torsions in the INS spectrum of K(CH₃SO₃) to be clearly seen. Reducing the size of the INS spectrum of K(CH₃SO₃) to fit on Figure 4, would mean that the modes at >400 cm⁻¹ would be very small.

We have combined Figures 6 and 7 and shifted the observed and calculated spectra further apart as suggested.

The long table, Table 3, is one of the major outputs of this work, so we believe it should be in the manuscript and not in the supplementary information.

4. The authors mentioned for the Li salt they observed distinct patterns. It's not the case for K salt since it's a purely ionic material. Have the authors tried to synthesis K salt and compare?

I have to say that I do not understand this comment. The K salt was purchased from Aldrich and has a stated purity of 98%. This was used for the single crystal X-ray structure determination and there was no evidence for impurities or a second phase, thus we did not see the need to prepare the K salt ourselves. We note that the infrared spectra of all the methanesulfonates, including that of the K salt, are shown and discussed in what was Figure 9 (now Figure 8).

Reviewer: 3

Comments to the Author(s)

Parker et al. report in this manuscript the study on the crystal structures of the lithium and potassium methanesulfonates. Both compounds have been characterised through the inelastic neutron scattering, Raman and infrared spectroscopies. In addition, DFT calculations have been performed. These studies are of high quality, but some doubts arise from this work.

The introduction is extremely short,

As noted above, we have expanded on the motivation for the paper and its relationship to previous work in the Introduction.

the syntheses of the studied compounds have not been reported in detail (what about the yield of the synthetic procedure and the analysis of the purity and homogeneity of the samples?),

As requested, we have provided more details on the synthesis and characterisation of the Li salts. and some parts of the crystallographic data have been communicated previously in conference abstracts and in a Thesis (in the case of the Li compound).

This is correct, but crucially the structures have not been reported by us, they have been determined completely from new. Also, as we state in the paper, the structures are not available from any recognised crystallographic database. As part of the submission process, the structures were deposited with the Cambridge Structural Database (CCDC 1989314 for $K(\text{CH}_3\text{SO}_3)$ and CCDC 1989315 for $\text{Li}(\text{CH}_3\text{SO}_3)$), thus are now generally available. This is new information.

In my opinion, in its present form, the manuscript is neither suitable nor scientifically sound.

Unsurprisingly, we completely disagree with this conclusion. We note that, to a degree, the referee has contradicted themselves: in their opening paragraph they state "These studies are of high quality", this cannot be "nor scientifically sound".

For these reasons, I do not recommend its publication in the journal RSOS.

We hope that with the improvements to the manuscript prompted by the reviewers, it will now be considered acceptable for publication in Royal Society Open Science.

Appendix B

ISIS Facility

Science and Technology Facilities Council
Rutherford Appleton Laboratory, Harwell Science and
Innovation Campus, Didcot, OX11 0QX United Kingdom

www.scitech.ac.uk

31st May 2020

Prof Stewart F. Parker
ISIS Facility
STFC Rutherford Appleton Laboratory
Chilton, Didcot
Oxfordshire
OX11 0QX
United Kingdom

Direct line +44 (0)1235 6182
E-mail stewart.parker@stfc.ac.uk

**“Structure and vibrational spectroscopy of lithium and potassium methanesulfonates”
(Manuscript ID: RSOS-200776)**

Dear Dr Smith,

Thank you for the reviewers comments, I am very pleased that both have now recommended publication in *Royal Society Open Science*. The only change to the manuscript is in “section 2.1. Materials” as Reviewer 3 requested that a more accurate value for the yield of the lithium salts was provided as (“...>90%” is not valid for me). I have rechecked my notes and amended the value to 96%.

Yours sincerely

Prof Stewart F. Parker